

# ProminTools: shedding light on proteins of unknown function in biomineralization with user friendly tools illustrated using mollusc shell matrix protein sequences

Alastair W. Skeffington and  Andreas Donath

Max Planck Institute of Molecular Plant Physiology, Potsdam, Germany

## ABSTRACT

Biominerals are crucial to the fitness of many organism and studies of the mechanisms of biomineralization are driving research into novel materials. Biomineralization is generally controlled by a matrix of organic molecules including proteins, so proteomic studies of biominerals are important for understanding biomineralization mechanisms. Many such studies identify large numbers of proteins of unknown function, which are often of low sequence complexity and biased in their amino acid composition. A lack of user-friendly tools to find patterns in such sequences and robustly analyse their statistical properties relative to the background proteome means that they are often neglected in follow-up studies. Here we present ProminTools, a user-friendly package for comparison of two sets of protein sequences in terms of their global properties and motif content. Outputs include data tables, graphical summaries in an html file and an R-script as a starting point for data-set specific visualizations. We demonstrate the utility of ProminTools using a previously published shell matrix proteome of the giant limpet *Lottia gigantea*.

# INTRODUCTION

Mineralized structures are formed by many organisms across the tree of life including bacteria, metazoans, plants and algae (*Skinner & Jahren, 2007*). These biominerals are critical for fitness, playing roles in support, defence, buoyancy, regulation of ion budgets and orientation, among others. Proteins have been found to be associated with many biominerals, and are hypothesised to have a key role in mineral synthesis (*Evans, 2019a*; *Evans, 2019b*; *Wang & Nilsen-Hamilton, 2012*). In some cases the roles of such proteins is relatively well understood, and some of the best studies examples come from molluscs (*Song et al., 2019*). For example, the proteolytic products of the Pif protein in molluscs have been shown to bind $CaCO_3$ crystals and induce formation of the aragonite polymorph of $CaCO_3$ in vitro (*Suzuki et al., 2009*). Knock-down of the Pif gene results in disordered

Corresponding author
Alastair W. Skeffington,
skeffington@mpimp-golm.mpg.de

growth of the aragonite crystals in the nacreous layer of the shell. In other systems, well studies examples include Amelogenin from tooth enamel, Silicatein from sponge spicules and Mms6 from magnetosome synthesising bacteria (*Wang & Nilsen-Hamilton, 2012*). However in the majority of cases the function of biomineral associated proteins remains elusive.

A common workflow in biomineralization research is to first clean a mineral preparation using detergents or oxidizing agents to remove loosely associated organic matter, and subsequently to dissolve the mineral, releasing tightly mineral-associate proteins into solution that can then be analysed using proteomic methods (*Marie et al., 2013b*). It is generally hypothesised that these proteins are likely to be involved in mineralization, and that proximity to the site of mineralization results in their incorporation into the mineral as it grows. Some of the proteins identified may have homology to proteins of known function or recognisable domains strongly suggestive of a certain function. For example, carbonic anhydrases have been found associated with calcium carbonate minerals in several organisms (*Le Roy et al., 2014*) and may aid generation of bicarbonate as a substrate for calcification. However there are generally many proteins in such data sets which lack similarity to proteins of known function (e.g., *Jackson et al., 2015*; *Kotzsch et al., 2016*; *Mann, Macek & Olsen, 2006*). Intriguingly, these proteins of unknown function often display unusual primary sequence characteristics, such as low complexity, biased composition and a high degree of predicted intrinsic disorder.

Informatic tools which allow biologists to easily investigate the global features of groups of proteins of unknown function relative to the background proteome are currently lacking. Thus many studies restrict their analysis of these proteins to noting the compositional biases or motifs which are obvious from manual inspection of the protein sequences. This method has the risk that important patterns in the data are missed and that rules are not applied consistently in identifying these patterns. Ideally the context of the proteome as a whole should also be taken into account. The more specific a feature is to the proteins of interest (POIs) the more likely it is to be involved in the specific function of those proteins. This notion is based on the well-established biological principle that the primary sequence of a protein is a strong determinant of molecular function, and that proteins with similar functions tend to share regions of sequence similarity. Thus, sequence motifs shared by a group of biomineral associated proteins are more likely to be involved in the specific function of those proteins if they are rare in the background proteome than if they are commonly found motifs. This principle is already used in various sequence analysis tools, including those seeking to identify important motifs (*Wagih et al., 2016*).

Although there are many tools available that allow researchers to investigate the properties of protein sequences in silico, including analysis of compositional bias, sequence complexity, intrinsic disorder and sequence motifs, such tools are not always easy to use. Some require command line use, data input formats differ, some can only run on one protein at a time and most require post-processing of the output to format the data for statistical and graphical analyses in commonly used environments such as Microsoft® Excel® or R. These tools also rarely allow researchers to compare two sets of sequences.

Here we present ProminTools, a set of easy-to-use tools for the statistical comparison of two sets of protein sequences, available as apps in the CyVerse Discovery Environment (https://de.cyverse.org/) (*Merchant et al., 2016*) or to run locally from a Docker<sup>TM</sup> container. The inputs are simply two fasta files containing the proteins of interest (POIs) and the background proteome respectively, while the outputs include data tables and an html document containing graphical summaries of the data and interactive tables for data exploration. To demonstrate the utility of these tools, we reanalyse a published data set of shell matrix proteins (SMPs) from the giant limit *Lottia gigantea* (*Mann & Edsinger, 2014*).

## MATERIALS AND METHODS

### ProminTools structure

The inputs for ProminTools are two fasta formatted files: the first containing the protein set of interest (POI set), and the second the reference or background proteins (typically the predicted proteome of the organism of interest). The background proteins are used for statistical comparisons with the POI sets, allowing the user the answer the following question: 'Are the features observe in the POI common in the background sequences or are they unusual?'.

ProminTools has two component programmes: "Protein Motif Finder" and "Sequence Properties Analyzer". Both are written in Perl and R and bundled with all dependencies in Docker<sup>TM</sup> (http://www.docker.com) containers. They can be run from Apps within the CyVerse Discovery Environment (*Merchant et al., 2016*) or on a personal computer via Docker<sup>TM</sup> Desktop. The primary outputs of the tools are data tables summarising key information from the comparison of the two sequences sets. The tools use these tables to generate an html file with a graphical summary of the information along with explanations, statistical analyses and interactive versions of certain data tables. A publication ready SVG (Scalable Vector Graphic) formatted figure is also generated by Protein Motif Finder. The R script that generates the html file from the data tables is also an output of the tools, allowing the user to reproduce the figures in the html report and to provide a starting point for further analyses specific to the data set. For licence information for all components of ProminTools the reader is referred to Data S1.

### Analyses performed by "Protein Motif Finder"
#### Motif finding with motif-x

Protein Motif Finder uses the motif-x engine (*Schwartz & Gygi, 2005*; *Wagih et al., 2016*) for motif finding. This engine was chosen because it breaks sequences down into their constituent motifs, by an iterative procedure that avoids oversimplification of motifs and prioritises motifs that are most enriched relative to a background sequence set. It is exhaustive for a given *p*-value and generates definite motifs rather than a position weight matrix, which simplifies downstream analyses and is more useful to molecular biologists. In this work it was always run with the recommended, conservative, binomial *p*-value of $10^{-6}$, but this parameter is user customisable in Protein Motif Finder. The motif width is also user customisable, while the minimum occurrences parameter is hard coded at

a value of five. Motif-x is run via the R module rmotif-x, centred on each amino acid sequentially and the results are combined. This procedure means that some motifs are likely to be redundant. For example, if the central residue is 'S' and the motif width 7 then the motifs '…S.S..' and '..S.S…(where dot represents any amino acid) may both be identified, but these would be collapsed to the single motif 'S.S'. Note that this procedure is conservative with respect to the original *p*-value calculated by motif-x. Significant motifs are then enumerated in the POI and the background sequence sets, and motif counts and enrichments reported in the output tables. Downstream analyses do not rely on the motif-x *p*-value, but only on calculated enrichment values for the motifs.

### Graphical representations of motif data

To provide a visual summary of the motif data, the motifs are represented in three wordclouds in the Protein Motif Finder output, which take into account two distinct measures of 'importance'. The first is the number of proteins in which that motif is found. The more proteins containing the motif, the more likely it is to have general importance in the function of the group of proteins. The second measure is the enrichment of the motif. The more enriched the motif the more unusual it is and thus is more likely to be involved in the specific function of these sets of proteins. A third measure attempts to combine the previous two by scaling them equivalently and then taking the product of the scaled values (PS-value). This measure prioritises motifs that are both highly enriched and found in a high proportion of the proteins.

In the output, proteins that are biased in sequence composition are also clustered based on their motif number and motif enrichment. The distance measure for clustering was calculated as one minus the Distance Correlation (*Szekely & Rizzo, 2013*) for all pairwise combinations of proteins or motifs, since this method is especially robust to outliers and produces reasonable results across a variety of datasets. Hierarchical clustering was performed using the Ward.D method. In Heatmap 1, the following filters are applied to select proteins and motifs for clustering: (1) Select proteins that contain a biased region (fLPS *p*-value $<10^{-20}$; user adjustable), (2) Remove infinitely enriched motifs. (3) Remove motifs not present in at least 3 POI proteins (if >10 proteins in POI set). (4) Select the 70 overall most enriched motifs. (5) Select proteins with a good motif based correlation to at least one other protein (dcor > 0.65, user adjustable). The filters for Heatmap 2 are the same except that filter 3 is not applied. The same protein and motif set is displayed in Heatmap 3 as in Heatmap 2 except that motif count is displayed instead of motif enrichment.

## Analyses performed by "Sequence Properties Analyzer"

Sequence Properties Analyzer performs the following analyses:

### Amino acid enrichment

Compositional bias is analysed using fLPS (*Harrison, 2017*) and the results collated to several files described in the html output of the program.

### Significance of sequence bias

To estimate the probability of obtaining the observed bias in amino acid composition in the POI set by random sampling of the background proteome, the following procedure

was implemented. First the degree of bias was quantified by calculating a bias index (BI):

$$BI = \sum_{Amino\ acids} \sqrt{\left(POIf\ req. - Proteome\ freq.\right)^2}$$

Where *POI freq.* is the frequency of the amino acid in the POI proteins, while *Proteome freq.* is the frequency of the amino acid in the background sequence set. The *BI* is calculated for 1,000 random samples of the background sequence set, each containing the same number of sequences as the POI set. A kernel density estimate of the distribution of *BI* is calculated, and a function approximating this distribution is generated. The area under the curve greater than the *BI* value of the POI set is used as an estimate of the probability of obtaining a sequence set of this degree of bias by chance, given this particular background proteome.

The only program we are aware of that makes a similar calculation is Composition Profiler (Vacic et al. 2007). However this makes the assumption that all POI sequences come from the same underlying distribution of amino acid frequencies and tests whether this distribution is significantly different from the background. ProminTools does not make this assumption, but accepts that the POI set my contain proteins with different types of bias, and thus analyses bias *per se* without reference to the type of bias.

### Sequence complexity

The program SEG (*Wootton & Federhen, 1993*) is used to identify low complexity regions in the datasets using default parameters, although these are customisable by the user in Sequence Properties Analyzer. For each protein, the percentage of the sequence identified as low complexity is calculated, and a Wilcoxon rank sum test with continuity correction is used to test whether there is a significant difference in the distribution of this percentage length between the POI and the background sequence set.

### Intrinsic disorder

Predicted intrinsic disorder was calculated using the VSL2 predictor (*Peng et al., 2006*), due to its speed and good accuracy (*Nielsen & Mulder, 2019*). This is the most time consuming step of Protein Sequence Analyzer and is thus parallelized in the implementation. For each protein, the percentage of the sequence identified as intrinsically disordered is calculated and a Wilcoxon rank sum test with continuity correction is used to test whether there is a significant difference in the distribution of this percentage length between the POI and the background sequence set.

### Charged clusters

Clusters of charged amino acids are identified using the SAPS software (*Brendel et al., 1992*).

## Data and methods for validation of ProminTools

Representative CxxC Zn finger proteins were chosen from the Wingender database (*Wingender, Schoeps & Dönitz, 2013*) and compared to the human proteome Swissprot database accessed on the 28/05/20. For the analysis of human low complexity proteins, all models were downloaded from Ensemble version 100. Models shorter than 100 amino

acids were removed, as were models with internal stop codons, resulting in 89562 proteins that were used as the background sequence in the analysis. The foreground sequence set was the 500 most biased proteins identified using fLPS (*Harrison, 2017*). These proteins were annotated using eggNOG mapper (*Huerta-Cepas et al., 2017*; *Huerta-Cepas et al., 2019*) with parameters "taxonomic scope hominidae, -target_orthologs all –seed_ortholog_evalue 0.001 –seed_ortholog_score 60 –query-cover 20".

### *L. gigantea* shell matrix proteome data

To illustrate the utility of the ProminTools package, we used the shell matrix proteome of the giant limpet, *L. gigantea* as published by *Mann & Edsinger (2014)* which is a reanalysis of their original data (*Mann, Edsinger-Gonzales & Mann, 2012*). The protein identifiers were extracted from Table S1 of (*Mann & Edsinger, 2014*) and the protein sequences extracted from files Lotgi1_GeneModels_AllModels_20070424_aa.fasta and Lotgi1_GeneModels_FilteredModels1_aa.fasta which were downloaded from the JGI (https://mycocosm.jgi.doe.gov/Lotgi1/Lotgi1.home.html) on the 5/02/2020. The final set of proteins consisted of 381 sequences, and are available in Data S2. This is one less than the number of accepted identifications in (*Mann & Edsinger, 2014*) since the protein Lotgi|172500 was not available in any database.

### Analyses of *L. gigantea* data using ProminTools

ProminTools was run locally using the shell matrix proteins as the foreground sequences and the 'Lotgi1_GeneModels_FilteredModels1_aa.fasta' file as the background proteome. The filtered models were chosen as they were considered likely to be a closer representation of the true proteome of *L. gigantea* than the 'All Models' set, and thus the more appropriate set for statistical comparisons.

### Additional analyses

Proteins were clustered based on motif content as an output of the Protein Motif Finder tool. To determine an optimal cluster number, manual inspection of a plot of the cindex (*Hubert & Levin, 1976*) for cluster sizes 2–50 was carried out. A cluster number of 8 seemed appropriate for the present work, since it captured the major patterns in the data without becoming too granular. These clusters were the input for further runs of Protein Motif Finder.

Sequence similarity was quantified using and all vs all pairwise BLASTp analysis, reporting the percentage identity of the top scoring high scoring pair (HSP), after applying an e-value cut-off of 0.01 and a cut-off specifying that the HSP alignment length must be at least 20% of the query length.

## RESULTS

### ProminTools provides a user-friendly method to analyse biomineralization proteomes

The Docker^TM image containing ProminTools can be run via a GUI on the CyVerse Discovery Environment without any need for use of the command line. Runtimes on CyVerse are variable due to variable resource availability, but a typical analysis with either

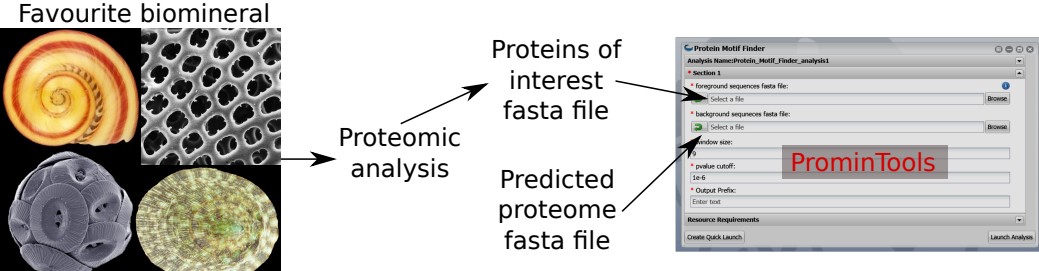

**Figure 1 Summary of ProminTools.** Proteomic datasets derived from analysis of biomineralizing organisms result in a fasta file containg the POI set which, along with the fasta file of the background proteome, make up the inputs for the ProminTools apps. The graphical interface shown is from the Cyverse Discovery Envronment (reproduced with permission). The outputs of the tools are detailed at the bottom of the figure. Attributions for the biomineral images are as follows: top left—*Vittina waigiensis* by H Zell (licence: CC BY-SA 3.0), top right—Radiolarian skeleton by Hannes Grobe (licence: CC BY 3.0), bottom left—*Coccolithus pelagicus* by Richard Lampitt and Jeremy Young, The Natural History Museum, London (licence: CC BY 2.5), bottom right—*Lottia mesoleuca* by H Zell (licence: CC BY-SA 3.0).

Protein Motif Finder or Sequence Properties Analyzer takes between 30 and 120 min to complete. Although ProminTools is designed to run in a Unix environment, it can also be run on a windows PC via Docker™ Desktop with simple commands in Windows® Power Shell™ (for details see https://github.com/skeffington/Promin-tools). On a Window® 10 machine with an Intel® Core™ i7-2600 3.4 GHz processer and 16 GB RAM, Protein Motif Finder completed analysing the *L. gigantea* data set in 11 min 30 s provided with 1 core and 2.5 GB RAM, while the Sequence Properties Analyser completed in 32 min 2 s provided with 5 cores and 5 GB RAM. The ProminTools workflow is summarised in Fig. 1.

## Validation of ProminTools
### Suitability for a range of data inputs

To ensure stability and good performance, we have tested ProminTools on a number of published and unpublished biomineralization datasets and used synthetic data to ensure that the program deals sensibly with unusual situations, such as small numbers of POI sequences or no motifs being found. An example analysis of a second data set, of proteins from freshwater mussel shells (*Marie et al., 2017*), is provided in Data S3.

### Validation with negative control protein sets

Five sets of 100 proteins were drawn at random from the *L. gigantea* proteome and each used as the POI set to run ProminTools in five separate analyses. No enriched motifs were reported in any of the analyses. In all analysis, there were no significant differences in the degree of sequence bias, sequence complexity or intrinsic disorder between the random 'POI' set and the background proteome. Representative analyses are provided in Data S4.
### *Validating motif retrieval in Protein Motif Finder*

Motif finding in PromintTools relies on the motif-x motif finding engine, which has already been well validated (*Schwartz & Gygi, 2005*). However to ensure that there were no bugs in our post-processing of the output we spiked motifs at known frequencies into a set of protein sequences and ran Protein Motif Finder with these sequences as foreground, and the un-spiked sequences as background. The spiked motifs were recovered at the expected frequencies.

We also validated Protein Motif Finder on a groups of sequences containing motifs that have already been established as important for protein function. For example CXXC class zinc finger factors are transcription factors and histone methyltransferases that bind to CpG elements via zinc fingers. The Zn binding residues consist of cysteines arranged in CGxCxxC motifs (*Wingender, Schoeps & Dönitz, 2013*). Using a set of CXXC zinc fingers factors as the POI set and the human proteome as the background set, Protein Motif Finder correctly identified CG.C..C as the most important motif, and found it to be 194 fold enriched in these proteins relative to the background proteome (Data S5).

It should be noted that not all motifs important for a protein's function will be enriched relative to the background. For example the L..LL motif is important in protein-protein interactions that regulate transcription (*Plevin, Mills & Ikura, 2005*). Using the 55 Swissprot proteins annotated as possessing an L..LL motif as the POI set and the Swissprot human proteome as background, Protein Motif Finder does not recover the L..LL motif (Data S6). This is because L..LL is relatively common in other contexts, and so is not significantly enriched in the POI set.

### *Validating the biological meaning of clustering by motif enrichment*

A key output of Protein Motif Finder is clustering of the POIs based on their motif enrichment. The usefulness of this clustering is based on the assumption that proteins within a cluster are likely to be involved in similar molecular processes. To test this assumption on a well annotated proteome, but focusing on biased sequences similar to those expected from biomineral associated proteins, we analysed the 500 most biased sequences from the human proteome. Of these, 303 could be annotated by eggNOG mapper (*Huerta-Cepas et al., 2017*) and they fell into 120 clusters when analysed with Protein Motif Finder (Data S7). Remarkably, proteins within a cluster all shared the same eggNOG functional annotation in 117 of the clusters, even when the proteins diverged significantly in primary sequence similarity as assessed by global alignments (Table 1). Of the input proteins, 53 were collagens, and 11 different types of collagen were successfully separated into separate clusters. The members of three clusters mapping to more than one annotation were clearly related within a cluster: two clusters contained two different types of collagen, while one cluster contained two types of epidermal growth factor-like domains.

## Global properties of *L. gigantea* shell matrix proteome

Previous analyses of the *L. gigantea* shell matrix proteome (*Mann, Edsinger-Gonzales & Mann, 2012*; *Mann & Edsinger, 2014*; *Marie et al., 2013a*) had noted a tendency for the

**Table 1** Annotations of the 10 largest clusters identified by ProminTools in an analysis of human 4 proteins of biased sequence composition.

| Cluster ID | Cluster size | Identity (%) in all-vs-all alignments (min; mean; max) | eggNog annotation | Proteins in cluster carrying this annotation (%) |
|---|---|---|---|---|
| 1 | 21 | 68.9; 87.7; 100 | Extracellular domain of unknown function in nidogen (entactin) and hypothetical proteins | 100 |
| 2 | 19 | 93.4; 96.9; 100 | Collagen type XI alpha 2 | 100 |
| 3 | 14 | 99.8; 99.9; 100 | Coiled-coil 2A | 100 |
| 4 | 13 | 76.2; 88.1; 100 | Collagen triple helix repeat (20 copies) | 100 |
| 30 | 5 | 99.6; 99.7; 100 | Nebulin repeat | 100 |
| 33 | 5 | 78.0; 86.6; 99.5 | Titin Z | 100 |
| 39 | 5 | 57.4; 74.5; 100 | Neuroblastoma breakpoint family member | 100 |
| 40 | 5 | 94.6; 97.8; 100 | WAS WASL interacting protein family member 1 | 100 |
| 41 | 5 | 67.9; 80.5; 95.9 | Golgin subfamily A member | 100 |
| 114 | 5 | 99.8; 99.9; 100 | Neurogenic locus notch homolog protein 4 | 100 |

proteins to be low complexity and disordered and that some proteins were enriched in particular residues. Here, ProminTools was used to put these observations on a more quantitative footing (Data S8, S9) and to discover enriched sequenced motifs in the data set from *Mann & Edsinger (2014)*, which contained 381 proteins. G rich motifs were found to be enriched most frequently among the proteins (Fig. 2A). Given that we are seeking to find the motifs that are shared within a group of proteins, Protein Motif Finder excludes motifs found in fewer than four proteins from certain plots to prevent the picture being dominated by a highly enriched motif found in very few proteins. The result can be seen in Fig. 2B, where Q containing motifs displayed the greatest enrichments relative to the background proteome. For example QQP was enriched 7.5 fold while Q.N.Q was enriched 6.1 fold (see data tables in Data S8 for these numbers). In general there is often a negative correlation between the number of proteins in a set containing motifs and the enrichment of those motifs. These two measures are combined (see Materials and Methods) in Fig. 2C, which emphasises motifs found in a high number of proteins with high enrichment (a high PS-value). For example, GG is found in 270 proteins and is 1.8 fold enriched, G..D in 234 proteins at 1.4 fold enrichment and NG in 249 proteins at 1.53 fold enrichment (Data S8).

The analysis of sequence bias and amino composition bias with Sequence Properties Analyzer was concordant with the motif finding results, in that G, P, Q and A were the most enriched amino acids (Fig. 2D). H, I, K, L, W, E and F were found to be the most depleted relative to the background proteome. The most commonly enriched amino acids were P, G, A and C (enriched in 87, 82, 74 and 69 proteins respectively, Fig. 2E). Amino acid residues C and A are not found among the most enriched motifs or the motifs with the highest PS-value, indicating that the proteins are sometimes enriched in an amino acid without that amino acid being embedded in a particular primary sequence context.

The shell matrix proteins showed a clear tendency to contain more low complexity sequence than the background proteome (Wilcoxon rank sum test, $p = 7.6 \times 10^{-6}$, Fig. 2F) but there was no significant tendency for the sequences to contain a greater proportion

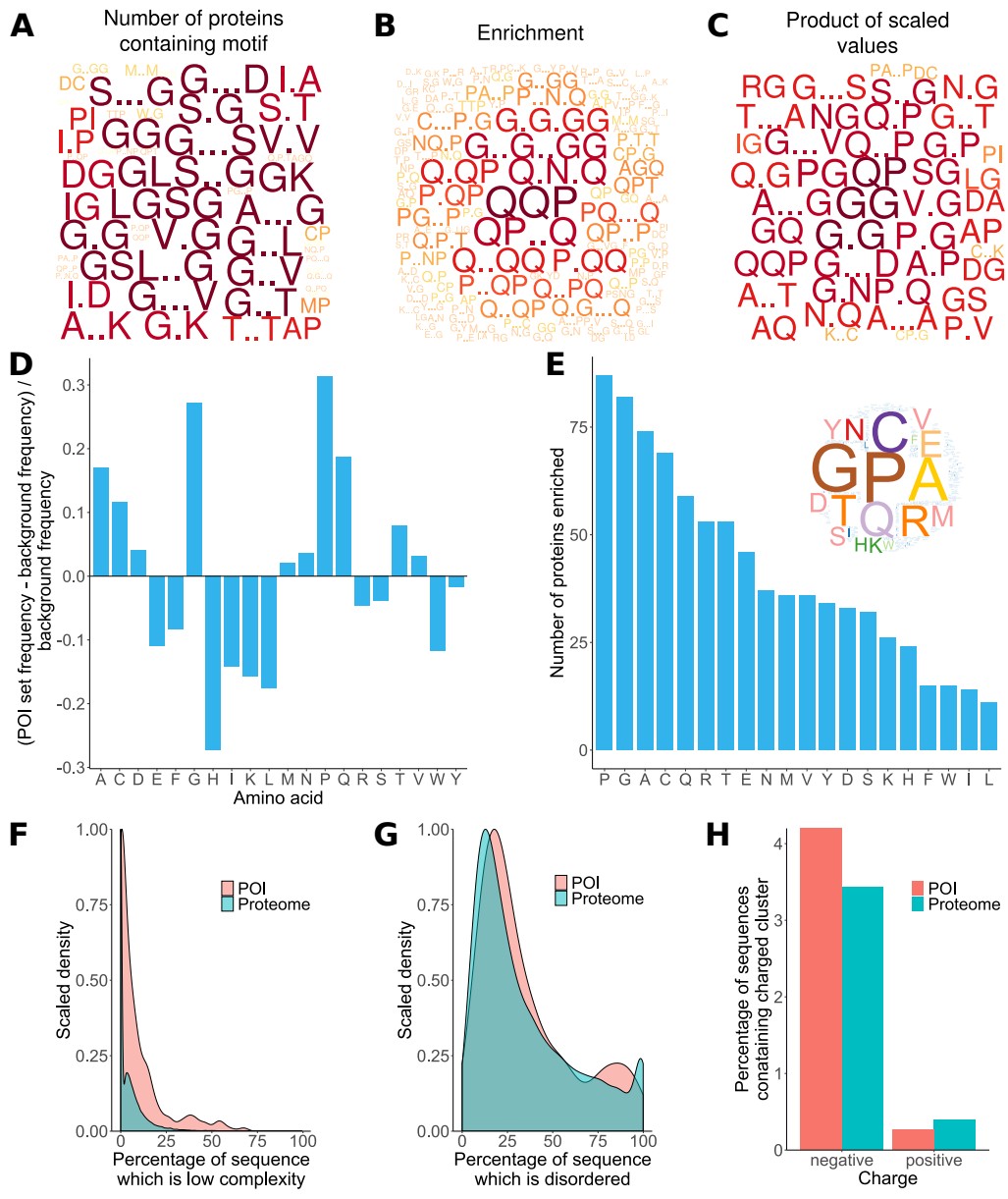

**Figure 2** **Properties of *L. gigantea* shell associated soluble proteome revealed by ProminTools.** Word-clouds are displayed where the height of the letter is proportional to: (A) the number of proteins in the SMP set containing the motifs, (B) the enrichment of the motif relative to the background proteome, and (C) the product of protein number and enrichment after scaling (PS-value). (D) The enrichment of amino acids in the SMP set relative to the background proteome. Values above zero indicate enrichment, and values below zero depletion. (E) The number of proteins in the SMP set enriched in each amino acid. Insert is a wordcloud summarizing the same data. (F) Density plot showing the distribution of the proportion of sequence length that is low complexity for the SMP proteins (labelled POI for Proteins Of Interest) and the background proteome. (G) Density plot showing the distribution of the proportion of sequence length that is predicted to be intrinsically disordered for the SMP proteins (POI) and the background proteome. (H) The proportion of sequences containing negatively and positively charged clusters of amino acids in the SMP proteins (POI) and the background proteome.

of predicted disordered sequence than the background proteome ($p = 0.1$). The shell matrix protein set contained a similar proportion of proteins with negative and positive clusters of amino acids to the background proteome (Fig. 2H).

### Clustering of proteins based on motif content reveals relationships not found by blast searches

The Sequence Properties Analyzer carries out three hierarchical clustering analyses (Data S8, 'Materials and Methods'). Eight protein clusters were identified (Fig. 3A), six of which contained more than two proteins. To investigate the nature of each cluster, Protein Motif Finder was re-run on each of the six main clusters (Data S10, Fig. 3B). Clusters 1 and 2 were rich in G containing motifs, especially NG.GG in cluster 1. Cluster 3 contained proteins rich in D containing motifs (especially D.NDD); cluster 4 in a variety of Q containing motifs; cluster 5 in C.I.P.D and C..YC..G and cluster 6 in various T and P containing motifs. By analysing the specific set of proteins in each cluster, the motifs identified are more specific to those proteins, and thus differ from the most enriched motifs in the data set as a whole displayed in Fig. 3A. For example D.NDD is the most prominent motif from the reanalysis of cluster 3, but is not among the most enriched motifs in the global analysis of the entire data set, demonstrating the value of this iterative approach.

Given the results of our validation analysis with human low complexity proteins, it can be hypothesised that proteins found within the same cluster have related function. Only one of the proteins (Lotgi1|143247, cluster 5) has an annotation: a 'four disulphide core domain protein' (Pfam PF00095), suggesting that it may function as a protease inhibitor. Given the lack of annotations, it was not possible to further test the relationship between cluster membership and function using the *L. gigantea* data.

We next asked whether the motif clusters reflected larger scale sequence similarity between the proteins within a cluster. To this end, protein sequences in each cluster were subject to an all-vs-all pairwise BLASTp analysis, which is summarised in the matrices in Fig. 3C for six of the clusters. In general larger scale sequence similarity was low within clusters, with only three protein pairs from the five clusters displaying identity above 50% for the highest scoring HSP. This demonstrates that the clustering method can be used to find similarities that are not obvious from BLAST searches.

## DISCUSSION

Proteins are a prominent part of the organic matrices of many biominerals and are thought to have a number of roles including catalysis, templating, and control of nucleation and crystal growth. Studies of biomineral associated proteins understandably often emphasise proteins with conserved domains, which lend themselves to discussions of their possible molecular functions. However most studies also identify many proteins of unknown function, many of which appear to be low complexity in nature, with biased compositions and a high proportion of intrinsic disorder. Although authors often carefully inspect their protein sequences and note sequences that appear particularly rich in certain residues or motifs, and note the degree of disorder, this information is rarely put in the context of the predicted proteome as a whole.
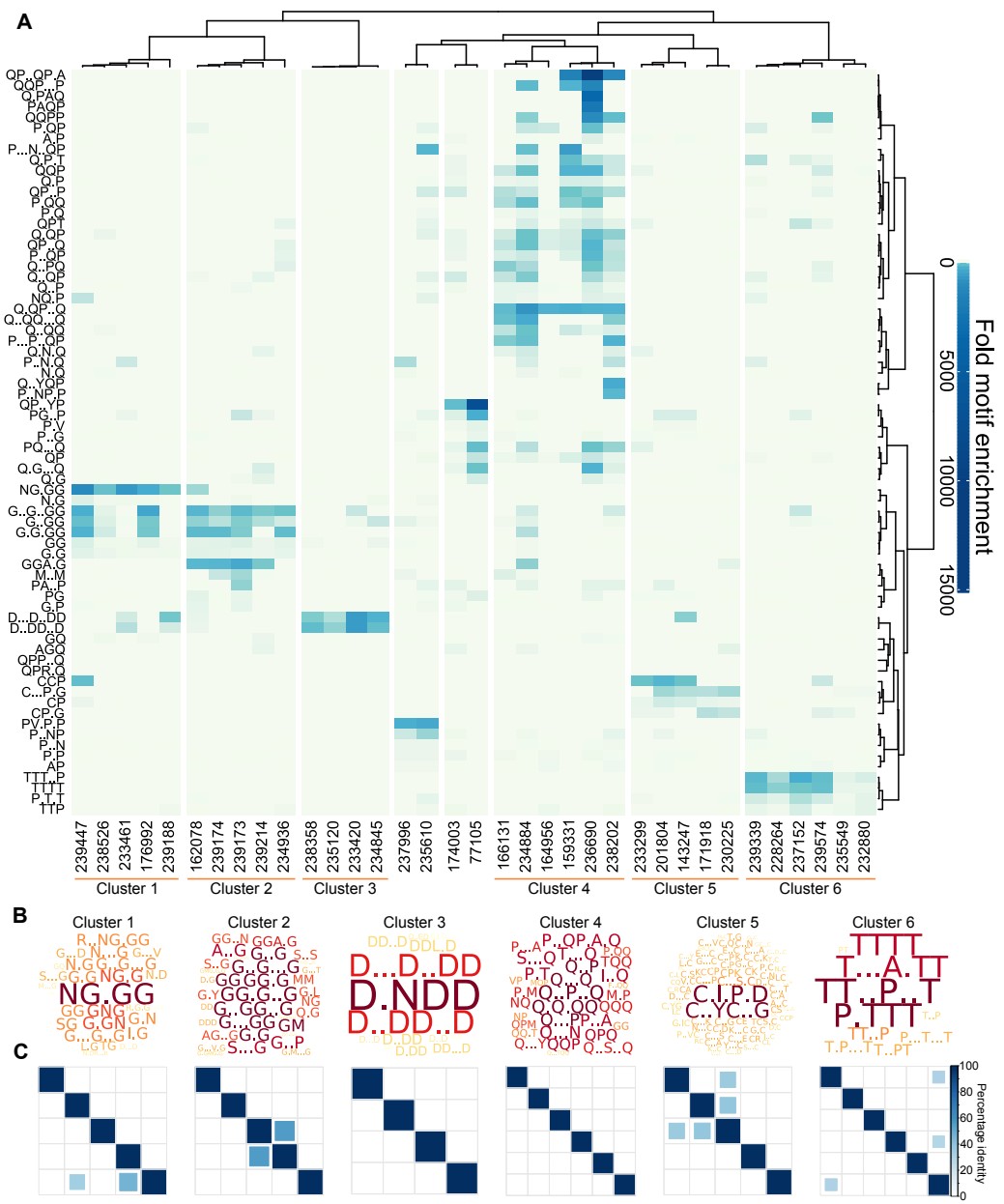

**Figure 3** *L. gigantea* **shell matrix proteins can be clustered based on motif content despite low sequence identity.** *L. gigantea* shell matrix proteins with biased composition can be clustered based on motif content despite low sequence identity. The heatmap displays (A) motif enrichment in the SMP set relative to the background proteome. Proteins are clustered by their motif enrichment pattern and motifs are clustered by their distribution amongst the proteins. Each motif is a row in the heatmap and each protein is a vertical column. For clusters 1–6, a wordcloud (B) representing the PS-value of the enriched motifs are displayed in addition to a heatmap (C) representing the percentage identity between all pairs of proteins in the cluster in an all-vs-all blastp analysis (see 'Materials and Methods').

Here we introduce ProminTools, a user friendly package that allows researchers to glean more information from primary sequences of proteins of unknown function and put this in the context of the background proteome. Importantly, ProminTools allows users with minimal bioinformatic skills to run a suite of analyses and produce visualization that would otherwise require a lot of scripting. The giant limpet *L. gigantea* has a complex shell matrix proteome for which two different data sets exist. The data analysed in the present study derives from all shell layers (*Mann, Edsinger-Gonzales & Mann, 2012*; *Mann & Edsinger, 2014*), and is thus more complex than the second data set that is derived from the aragonite shell layers only, excluding the calcitic layers (*Marie et al., 2013a*).

ProminTools revealed a complex array of strongly enriched motifs in the *Mann & Edsinger (2014)* data set, which were not uncovered in the original study. Q, P and G rich motifs were particularly prominent and the proteins could be clustered based on their motif content even when they shared little larger scale sequence similarity. Re-running Protein Motif Finder on each of these clusters revealed unique motif profiles that could be hypothesised to be important for the molecular function of proteins in the group. For example, one group was enriched in acidic (D rich) motifs, another in Q and P rich motifs and other in G rich motifs. Interestingly, the Marie et al. study also identify a group of low complexity proteins rich in Q, suggesting that the functions of these proteins could be important for formation of all shell layers or just the aragonite layers, but that they are unlikely to be specific to the calcite layers.

We hypothesise that clustering protein sequences with biased composition based on their motif enrichment patterns can be used to group proteins of related function. Although this hypothesis has yet to be confirmed on biomineral associated protein data sets, we show that this procedure can group functionally related proteins of biased composition from humans. Additional support for the idea comes from a previous study in which accurate predictors of enzyme function were built using the motif content of protein sequences (*Ben-Hur & Brutlag, 2006*).

We found that the shell matrix proteins as a group were significantly lower in complexity than the background proteome, providing a statistical underpinning for this observation, and supporting the conclusion of *Marie et al. (2013a)* who noted the high proportion of low complexity sequences in their data set. The Mann et al. studies (*Mann, Edsinger-Gonzales & Mann, 2012*; *Mann & Edsinger, 2014*) highlight several proteins in their data which have high degrees of intrinsic disorder. Here, using the Sequence Properties Analyzer we were able to demonstrate that this is not a general feature of the data set, which is not predicted to be significantly more disordered than the background proteome. This highlights the importance of the proteome context when assigning significance to protein features, and demonstrates that the generally observed correlation between protein disorder and low complexity (*Mier et al., 2019*) does not hold in every data set.

The role of low complexity regions in biomineralization has only been determined in a very few cases. For example, the enamel protein Amelogenin has a central block of hydrophobic sequence rich in P, H and Q. Intramolecular hydrophobic interactions involving this regions are thought to be critical for self-assembly of Amelogenin into

nanospheres and higher order structures that regulated crystal growth (*Wang & Nilsen-Hamilton, 2012*). It is possible that the Q and P rich regions in the *L. gigantea* shell matrix proteins might have a similar role in driving self-assembly processes.

Although at present we can only speculate on the role of low complexity proteins in biomineralization, it is clear that low complexity sequences are not unique to biomineralization related proteins. Depending on the species, 22–36% of residues in eukaryotic proteins fall into low complexity regions (*Wootton, 1994*). It remains to be investigated whether the low complexity regions of biomineralization related proteins have features that set them apart from other low complexity regions in proteomes, and ProminTools could be used to investigate such questions.

ProminTools allows researchers to easily find patterns in their data, but it has limitations and judgement should be applied in interpreting the output. For example, patterns found by ProminTools can reflect technical biases as well as biological signals. Post-translational modifications of particular residues could affect peptide detectability and thus protein inference, leading to biases in the input data. It should also be remembered that ProminTools is primarily a tool for hypothesis generation. For example, proteins which share similar motifs can be hypothesised to perform similar molecular functions, but this may or may not be the case for a particular biological system, and experimental validation is required. ProminTools will be at its most useful when combined with other methods for spotting repeating patterns in sequences (e.g., HhpreID *Zimmermann et al., 2018*, Meme *Bailey et al., 2009*) or simply inspecting dot-plots) and when put in the context of additional information such as known domain content, post-translational modifications, phylogenetic distributions and expression patterns.

We would like to point out that ProminTools can be used for any pairwise comparison of sets of protein sequences. For examples, protein sets associated with different part of a biomineral or different developmental stages could also be compared, and if carefully carried out, cross-species comparisons could also be made. The latter could be particularly useful, since the fast evolving nature of low complexity sequences (*McDougall, Aguilera & Degnan, 2013*) can make it difficult to detect homology. It could also be applied to other protein sets rich in low complexity sequences, such as proteins found in pathological amyloids associated with diseases such as Alzheimer's and Parkinson's (*Kumari et al., 2018*).

## CONCLUSIONS

ProminTools will help researchers generate new hypotheses about the important of particular motifs and protein chemistries in their system of interest and provide new directions for experimental work. Putting the patterns identified into the context of the rest of the proteome ensures that features that are genuinely overrepresented in the POIs are prioritised for further study.

## ACKNOWLEDGEMENTS

We gratefully acknowledge Dr. André Scheffel for advice and critical reading of the manuscript.

### Funding

This work was supported by a fellowship from the Alexander von Humboldt Foundation. The funders had no role in study design, data collection and analysis, decision to publish, or preparation of the manuscript.

### Grant Disclosures

The following grant information was disclosed by the authors:
Alexander von Humboldt Foundation.

### Competing Interests

The authors declare there are no competing interests.

### Author Contributions

- Alastair W. Skeffington conceived and designed the experiments, performed the experiments, analyzed the data, prepared figures and/or tables, authored or reviewed drafts of the paper, and approved the final draft.
- Andreas Donath performed the experiments, authored or reviewed drafts of the paper, containerised the tools described, and approved the final draft.

### Data Availability

A detailed description of ProminTools and instructions for use are available at GitHub: https://github.com/skeffington/Promin-tools.

The docker containers for ProminTools are available at Figshare:

- Skeffington, Alastair; Donath, Andreas (2020): Sequence Properties Analyzer Docker Image. figshare. Software. https://doi.org/10.6084/m9.figshare.12670817.v1.

- Skeffington, Alastair; Donath, Andreas (2020): Protein Motif Finder 04 Docker image. figshare. Software. https://doi.org/10.6084/m9.figshare.12667070.v1.

Further information on accessing ProminTools is available in Data S11.

### Supplemental Information

Supplemental information for this article can be found online at http://dx.doi.org/10.7717/peerj.9852#supplemental-information.

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
