# Peer review of "ProminTools: shedding light on proteins of unknown function in biomineralization with user friendly tools illustrated using mollusc shell matrix protein sequences"

_PeerJ, doi:10.7717/peerj.9852_

## Round 0.1 · original submission · Major Revisions

Dear Drs. Skeffington and Donath:

Thanks for submitting your manuscript to PeerJ. I have now received three independent reviews of your work, and as you will see, the reviewers raised some concerns about the research. Despite this, these reviewers are optimistic about your work and the potential impact it will have on research studying computational approaches to characterizing biomineralization proteins. Thus, I encourage you to revise your manuscript, accordingly, taking into account all of the concerns raised by all three reviewers.

There appear to be several concerns regarding your methodology, and specifically the computational approach. Please consider the need for lacking controls or other approaches for objective assessment of ProminTools.

Please note that Reviewer 2 kindly provided a marked-up version of your manuscript.

Therefore, I am recommending that you revise your manuscript, accordingly, taking into account all of the issues raised by the reviewers.

I look forward to seeing your revision, and thanks again for submitting your work to PeerJ.

Good luck with your revision,

-joe

·

Basic reporting

Present study claims taking advantage of using a protein sequence specific set of collection available in databases (limpet shell matrix proteins) to carry out a bioinformatics analysis aiming at providing an easy visualizing of protein sequences profile clustering.

The subject clearly matters in the context of biomineralization control. In fact many of the proteins recovered from proteomics methods in biomineralized biological systems (including mollusc shell), do not give rise to identification and functional characterization by searching for sequence similarities in general databases. The “unknown” status represents a significant fraction of the shell proteomes, in part due to the structure of these proteins. Thus, the development of meaningful molecular descriptions of these proteins remains a key challenge for now in different structural- ,functional-, evolutionary biology issues also for different prospects, e.g. biomimetics, …

Experimental design

The manuscript provides a detailed and illustrated operating mode of a user-frendly tool package designed to help expose patterns in a set of proteins of interest in comparison to an alternative set of proteins.

This is an effective and elegant way to do so since a molecular representation of the disordered state based on diverse sources of structural data may exhibit complex and very different averaging behavior which is presently a big deal in understanding biomineralization mechanism, in the context of biochemistry and evolutionary biology.

Validity of the findings

Though, there are several concerns in the manuscript, arising from unbalanced consideration of the biological context, limited and even oversimplified in some places.

This can be examplified early in the introduction section and proceeds along with the discussion:

l.15 – molluscs are definitely metazoans

l.33 – general references for proteins involved in biomineralization, as the backbones of present report assessments, may come from works prior to 2019, even 2012

l.35 – Pif protein in molluscs should be referred to with Suzuki et al. 2009 (instead of Song et al. 2019). The whole sentence containing this reference should be reworked to be tightly reframed with better focus.

l.39 onwards – Proteomics method (the highway to recover the dedicated protein sequences) is introduced with kind of workflow and upstream cleaning procedures (required for cautiousness) mix up. Irrelevant to the present work purpose, with regard to the latter. On the contrary, critical limits for sequence identification, e.g. amino acid composition, presence of multiple repeats and low-complexity domains, solubility properties, glycosylations / phosphorylations sites, cleavage sites availability, … or fragmentation techniques have been overlooked even though several issues have driven some packages of the tool.

Then in the ‘analysis’ (Motif Finder) section

- What is the guideline for providing two different ‘sets’ of protein sequences to compare the sequences of interest with? Do the authors take each set apart for the specific package application. This point should be clarified.

Careful attention should be paid by the author when writing :
- ‘Although the human eye is good at detecting patterns, this method has the risk that important patterns in the data are missed … .’(l.57)
- ‘Protein Motif Finder can facilitate researchers in detecting the important patterns in
their data’ (l. 267)
Oversimplification or statement driven deduction are both compelling reasons to ask for a step further discussion of these issues.

Minor issues :
A few spelling errors (e.g. l. 75, 195, 217, 329) indicate at least thorough editing is needed.

Additional comments

In conclusion, present manuscript reports for sure on original primary work as for the tool development.
Wether or not it falls within the scope of the journal is not within my discretion.

Reviewer 2 ·

Basic reporting

Please refer to the attached document.

Experimental design

Please refer to the attached document.

Validity of the findings

Please refer to the attached document.

Additional comments

Please refer to the attached document

Annotated reviews are not available for download in order to protect the identity of reviewers who chose to remain anonymous.

Reviewer 3 ·

Basic reporting

No comments.

Experimental design

The article presents a useful tool, which could compare two sets of protein sequences in terms of their global properties and motif content. It will surely help the user to discover the potential characteristics of interested proteins in a user-friendly way.
But as for the computing methods, I have several questions:

1. Are there any other useful tools which can be compared with your tools? And what are the performances of these tools and what are the advantages of ProminTools? You should provide strong evidence to show that ProminTools are in high accuracy and efficiency.

2. Another questions are about the test sets.
(1) Why do you use the proteome of the giant limpet? Given that there are two different results based on the same raw sets, how could you illustrate the superiority of ProminTools?
(2) Have you ever used any other data sets to prove the utility of ProminTools?
(3) How many shell matrix proteins do you select as the foreground sequences? I couldn't find the exact number only if I look back to the data in the paper of Mann 2014.
(4) Have you ever set negative control ground in the case of the giant limpet? A group of randomly selected proteins is strongly suggested.

3. As for the methods:
(1) In terms of the runtime, have you ever tried to run the scripts in parallel on the GPU? It may be helpful to decrease the runtimes in future.
(2) The function of low complexity proteins is surely a hard question. Have you ever machine learning algorithm to solve this problem?

Validity of the findings

ProminTools are designed to deal with the biomineral associated proteins, and the author also mentioned that the tool can also be used in many other cases. It seems to be a powerful tools, but I have one major questions:

The tool did provide the motif and sequence properties, but could the tool give a scoring report for POI set and potential shell matrix protein lists?It will be more useful for the investigators.

Additional comments

No more comments.

---

## Round 0.2 · accepted · Accept

Dear Drs. Skeffington and Donath:

Thanks for revising your manuscript based on the concerns raised by the reviewers. I now believe that your manuscript is suitable for publication. Congratulations! I look forward to seeing this work in print, and I anticipate it being an important resource for groups on research studying computational approaches to characterizing biomineralization proteins. Thanks again for choosing PeerJ to publish such important work.

Best,

-joe

Reviewer 2 ·

Basic reporting

The authors have improved the manucscript by taking into account my earlier comments. I recommend the article for publication without the need for further revision.

Experimental design

Not applicable

Validity of the findings

The authors have included new data set to demonstrate the usefulness of their workflow and valditate the tool.

Additional comments

The authors have improved the manucscript by taking into account my earlier comments. I recommend the article for publication without the need for further revision.